# Bioconjugation Strategies for Connecting Proteins to DNA-Linkers for Single-Molecule Force-Based Experiments

**DOI:** 10.3390/nano11092424

**Published:** 2021-09-17

**Authors:** Lyan M. van der Sleen, Katarzyna M. Tych

**Affiliations:** Groningen Biomolecular Sciences and Biotechnology Institute, University of Groningen, 9747 AG Groningen, The Netherlands; l.m.van.der.sleen@rug.nl

**Keywords:** optical tweezers, atomic force microscopy, single-molecule fluorescence, single-molecule force spectroscopy, unnatural amino acids, non-canonical amino acids, functionalization strategies

## Abstract

The mechanical properties of proteins can be studied with single molecule force spectroscopy (SMFS) using optical tweezers, atomic force microscopy and magnetic tweezers. It is common to utilize a flexible linker between the protein and trapped probe to exclude short-range interactions in SMFS experiments. One of the most prevalent linkers is DNA due to its well-defined properties, although attachment strategies between the DNA linker and protein or probe may vary. We will therefore provide a general overview of the currently existing non-covalent and covalent bioconjugation strategies to site-specifically conjugate DNA-linkers to the protein of interest. In the search for a standardized conjugation strategy, considerations include their mechanical properties in the context of SMFS, feasibility of site-directed labeling, labeling efficiency, and costs.

## 1. Introduction

Single molecule force spectroscopy (SMFS) has proven to be a powerful tool to investigate the properties of individual proteins, including mechanical stability [1], ligand-binding [2] and protein folding [3,4]. The three most commonly used methods are optical tweezers [4,5] (OT), atomic force microscopy [6,7] (AFM), and magnetic tweezers [8,9] (MT), which are able to measure forces in the piconewton range. While each method has its own features and limitations [10], it is typically required that the molecule under study is attached to a probe, i.e., an optically trapped bead, a tip of the AFM cantilever or a magnetic bead (Figure 1). Background signals can be reduced by excluding non-specific short-range interactions between a surface and the protein of interest (POI) using a flexible linker between the protein and the trapped probe [5]. For optical tweezers, the linker additionally prevents the POI from being located directly in the path of the laser beam, avoiding thermal denaturation and accelerated photo-damage by the laser.

The most commonly used linkers for AFM are polyethylene glycol (PEG) polymers due to their high flexibility and well-characterized elastic behavior. Such polymers are commercially available with a variety of terminal functional groups. However, PEG polymers have shown polydispersity and isomerization at tensions exceeding 300 pN [11]. When specifically considering the application of PEG linkers with MT, analysis may be complicated due to PEG binding to the magnetic beads [12]. More recently developed flexible linkers for SMFS studies are elastin-like polypeptides (ELPs) [13], consisting of a repetitive GXGVP motif (X is any amino acid except proline). ELPs are monodisperse and can readily be expressed with site-specific handles for conjugation [13]. Many more amino acid-based flexible linkers have been designed [14]; however, the vast majority have not yet been explored and still require characterization for SMFS experiments. On the other hand, DNA-linkers have a well-characterized contour length and behavior under force and are compatible with force-measurements of proteins. Even though applied forces are limited to forces below the DNA overstretching regime of approximately 65 pN at common lengths of ~500 bp [15], DNA-linkers are most commonly used in protein measurements with OT due to their predictable properties and low level of intrinsic Brownian fluctuation at high forces [12]. To conjugate the DNA-linker to the appropriate probe for OT measurements, silica or polystyrene beads coated with, for instance, (strept)avidin, anti-digoxigenin or azide-groups and biotinylated, digoxigenin- or alkyne-modified DNA primers are commercially available (e.g., from Spherotech, Lumicks, Biomers, and Jena Bioscience).

A general consideration of all three SMFS methods is the bioconjugation strategy of attaching the linker to a probe or a specific site on the POI. One should consider the applicability and feasibility of the strategy for each purpose. Other considerations are the cost, the compatibility with the protein, the size, and the strength of the complex. This review provides a general overview of bioconjugation strategies to connect the DNA-linkers with the POI. Their strengths and weaknesses for SMFS measurements will be discussed together with an outlook for strategies that may become more commonplace in the future.

## 2. Non-Covalent Molecular Handles

To date, non-covalent molecular handles are most often utilized in SMFS experiments. Strategies include the commercially available bioconjugation pairs streptavidin:biotin [16], Histidine-tag:NTA [17,18,19] (Figure 2a) and anti-digoxigenin:digoxigenin [20]. The experimental conditions for binding do not require additional catalysts and sufficient binding may be achieved within a few hours. However, strategies involving streptavidin and digoxigenin would require fusion of the protein to the POI and are therefore not practical in case of DNA-POI linking. Applications of these handles are therefore restricted to the attachment of linkers to the probes.

It should be noted that non-covalent molecular handles show rupture forces around 100–200 pN [18], depending on their loading rate [21]. Extremely mechanically stable proteins may exceed a rupture force of 500 pN. In such experiments, double-stranded DNA is typically replaced with PEG-based spacers, having the additional benefit of enabling DNA-interacting enzymes to be studied [22]. Studies of the unfolding of these mechanically stable proteins by SMFS would be very challenging with the aforementioned non-covalent handles due to this high rupture force. Cohesin:Dockerin type III and SdrG:Fgß are conjugation pairs that show extreme mechanical stability despite their non-covalent interactions with rupture forces exceeding 500 pN [22,23] (Figure 2a). Cohesin and Fgβ can be expressed as fusion proteins to the POI, although they are relatively large complexes and may affect the expression, solubility, and functioning of the POI. Although the two complexes provide a promising ligation strategy with extraordinarily strong non-covalent interactions, in practice it is a less elegant method; an additional method for the ligation of a protein (Dockerin or SdrG) to a DNA-linker is still required.

## 3. Covalent Molecular Handles

The rupture of a covalent bond can require forces exceeding 2 nN [24] and therefore provides a more stable and stronger link between the POI and DNA than the aforementioned methods. There is a wide range of strategies that can form covalent bonds, either requiring a short peptide-tag or a single mutation on the POI. These are described in the sub-sections below.

### 3.1. Covalent Peptide-Tags

Several short peptide sequences have been identified that form covalent bonds, either enzymatically or spontaneously [25,26,27]. Binding partners may be either other short peptides, coenzymes, small proteins or other moieties. These handles can be classified into three different categories: enzymatic linking/ligation, enzymatic self-labelling and isopeptide bonds [25].

Enzymatic linking/ligation tags, such as ybbR [28,29], LPXTG [30,31,32] and NGL [33,34] tags, all require an additional enzyme to link the short peptide sequence to another peptide or coenzyme (Figure 2b). The speed, reversibility, efficiency, activation conditions and requirement for metal ions of the enzyme should be taken in consideration when choosing the appropriate strategy (Table 1). Furthermore, the choice of the position of the tag should be considered. Whereas LPXTG and NGL tags can only be located at one of the termini of the POI, the ybbR-tag can be positioned more flexibly in the POI. The ybbR-tag additionally has the advantage that 3′ and 5′ CoA-modified oligonucleotides can readily be purchased while the introduction of short peptides requires additional conjugation steps. Enzymatic self-labelling strategies include the HaloTag [35,36,37] and hAGT/SNAP tag [38,39] (Figure 2c), which can be expressed as a fusion protein. These tags can bind to moieties containing chloroalkane or benzylguanine (BG), respectively, whereas only BG-modified oligonucleotides can readily be purchased.

Proteins containing isopeptide bonds have been engineered by splitting the full protein into two fragments, consisting of a peptide tag and a protein binding partner. This pair can rapidly and spontaneously form an irreversible bond. Examples are the SpyTag:SpyCatcher [43,44,45,46] and SnoopTag:SnoopCatcher [47] systems (Figure 2c). Difficulty arises when trying to purify the Catcher-fragment as an independent construct. For instance, while the SpyTag is fused with the POI, the SpyCatcher fragment is attached to the DNA-linker using an additional linker-protein: the maltose binding protein. This extra protein-fragment is required for convenient purification [44].

Although the covalent tags are significantly smaller in size than SdrG:Fgß, it should be noted that the tags may still interfere with the protein function. The loading geometry and possible unfolding [47] should also be carefully considered for these tags.

### 3.2. Strategies with Single Amino Acid Mutations

Conjugation strategies requiring only one amino acid have the advantage that all surface-accessible amino acids can be utilized for binding. As only the mutation of a single amino acid is needed, the function of the POI has the potential to be less affected than when introducing a peptide-tag, although functional tests should always be performed after mutation or insertion. Furthermore, the natural occurrence of the amino acid used as a target should be considered to prevent non-specific binding. Below, we describe different approaches to perform this type of attachment.

#### 3.2.1. Lysine- and Cysteine- Coupling Reactions

Lysine and cysteine are the most-targeted amino acids in proteins for labelling [48]. A wide array of lysine-targeted coupling strategies have been developed, of which EDC/NHS coupling reactions are the most common [49] (Figure 3a). Due to the relatively high abundance of lysine in proteins, site-specific labelling is often not possible. When targeting lysine, the pH should also be considered, as the involved reagents are optimally activated under acidic conditions. Additionally, possible side-reactions may occur with the N-terminal amino group and other amino acids such as serine and tyrosine when not working at the appropriate pH [50]. Cysteines are less abundant in nature and are mostly coupled with maleimides or thiols (Figure 3a). However, the introduction of cysteine-mutations in the POI may cause multimerization and aggregation, and therefore requires a reducing agent prior to coupling. Care should be taken with the pH and the removal of the reducing agent to prevent reactions with free amines such as lysine and to avoid unreactive maleimides. The optimal pH should range between 6.5 and 7.5 for a chemoselective reaction with cysteines. Under more alkaline conditions, lysines are in competition to also react with maleimides [51]. An increase in labelling specificity may be achieved by using dinitroimidazoles for bioconjugation [52].

Although the linking efficiency may be relatively low, the experimental steps required for coupling to cysteines are straightforward. Moreover, maleimide modified oligonucleotides are commercially available, making this an accessible strategy to implement.

#### 3.2.2. Non-Canonical Amino Acids

Selective targeting of canonical amino acids for bioconjugation is not always possible due to their native occurrence in the POI. Although these native amino acids can be removed, the protein stability and function may be severely affected. An emerging field is the introduction of non-canonical amino acids (ncAAs). By introducing an orthogonal aminoacyl-tRNA synthase to the translational machinery of the cell, a wide variety of chemical handles can be introduced for site-specific conjugation [53] (Figure 3b). Compatible reactions include click chemistry, oxime ligation, and inverse electron demand Diels–Alder reactions [54] (Figure 3c). This technique can reliably be used in *E. coli*., though the number of orthogonal translation systems is limited for other cells, such as yeast [55] and mammalian cells [56].
Figure 3Conjugation strategies involving single amino acid mutations. (**a**) The native amino acids lysine (**left**) and cysteine (**right**) can be targeted by several strategies for conjugation. Lysine is naturally more abundant than cysteine. (**b**) Incorporation of non-natural amino acids (ncAAs) require orthogonal aminoacyl-tRNA synthetase (blue) to incorporate ncAAs (orange) at a stop-codon. Due to competition with endogenous RF1, truncated proteins are also found. (**c**) Several compatible reactions for DNA-protein bioconjugation involve ncAAs (TPPO: triphenylphosphine oxide). Rate constants (k) [55] are given in M^−1^s^−1^.
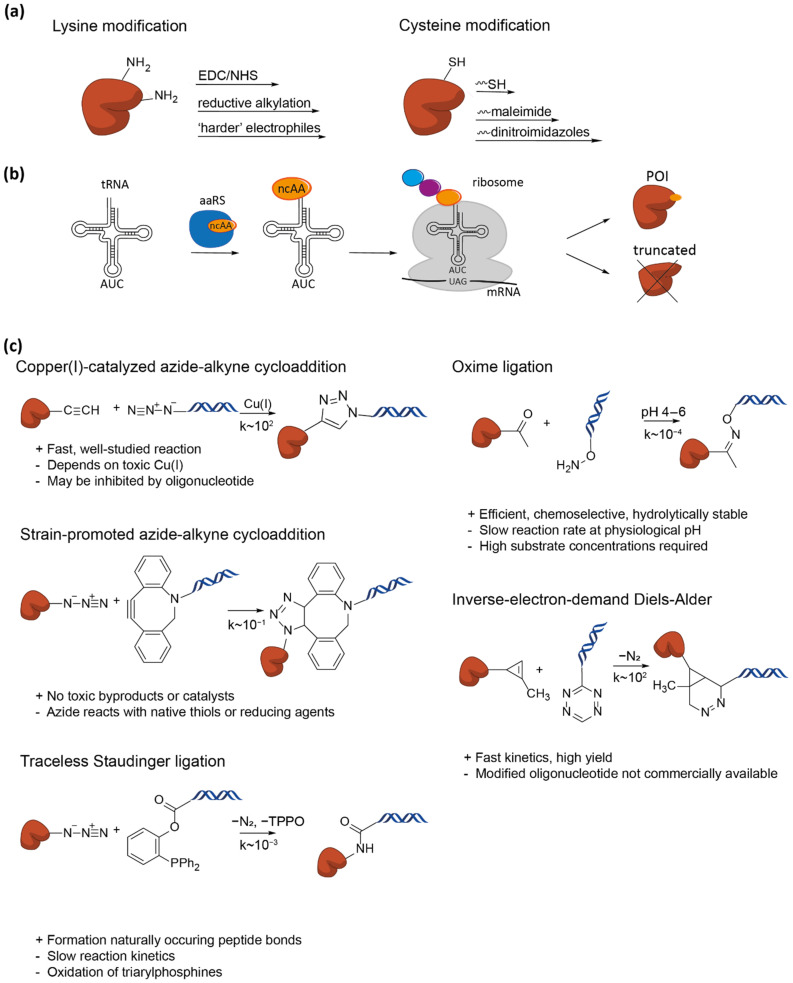


These conjugation-reactions can be performed as a fast one-pot reaction requiring straightforward experimental steps and can be performed under biological conditions. Click chemistry is one of the most widely used bioorthogonal strategies involving ncAAs. Due to its biocompatibility, its applications range from bio-imaging to the design of novel drug delivery systems. Classic examples of click chemistry are copper(I)-catalyzed cycloaddition between an azide and alkyne, and strain-promoted azide-alkyne cycloaddition, using strained alkynes such as DBCO. In general, a higher reaction rate can be achieved with copper(I)-catalyzed click chemistry [57], though conjugation of protein and DNA is possibly inhibited by chelation of Cu^2+^ by the oligonucleotide itself [58]. Strain-promoted cycloaddition avoids the requirement for metal ions due to the ring-strain-driven reaction and is an efficient way to introduce covalent linkages. However, moieties such as DBCO are expensive due to their low-yielding synthesis and the commercial availability of DBCO-conjugated oligonucleotides is limited.

Targeting ncAAs for bioconjugation is a versatile strategy that can be utilized for a broad range of proteins for SMFS. Its advantages are its independence of the amino acid content, their size, labelling efficiency, strength of the linker and the broad range of possible orthogonal reactions. Nonetheless, straightforward introduction of ncAAs in practice still has several bottlenecks. The orthogonal translational machinery has to be introduced to the cell, ncAAs have to be supplied to the medium in excess, and often the yield and the incorporation efficiency is low. Truncated proteins should therefore be removed by purification methods. Altogether, the use of ncAAs for bioconjugation is a very promising field; however, optimization of efficiency and the corresponding costs are required in order to outcompete other bioconjugation strategies.

### 3.3. General Considerations on the Fusion of Tags

While peptide tags can be fused with the POI using cloning strategies, DNA fused with the appropriate binding partner is not always commercially available. Implementation requires additional chemical steps or alternatives of which the behavior should be carefully examined under application of force. For instance, DNA functionalized with maleimides, azides and coenzyme A are commercially available. For other strategies, such as enzymatic linking by using an ybbR-tag, kits to functionalize the protein with DNA can be purchased. Other modifications, such as small peptides and DBCO, require an additional step to functionalize the DNA with the appropriate handle. Common strategies include cysteine-, click- and EDC/NHS chemistry, making use of DNA functionalized with a thiol, maleimide, azide or amine group.

Some methods require several manipulation steps and incubation of the POI before appropriate linkage is achieved. If the POI can, for instance, only be obtained in small quantities or is unstable, these methods may not be appropriate. Conjugation between isopeptides can be fairly easily achieved. Though, with the tag located on the POI, convenient purification of the Catcher-fragment requires fusion to another protein, such as the maltose binding protein [34].

## 4. Discussion

A wide variety of tools has been developed over recent decades to conjugate DNA to proteins for several purposes. Besides general considerations such as cost, compatibility, feasibility, and the ease of introducing the functional group, single molecule force spectroscopy demands linkers to have a sufficiently high rupture force and well-defined behavior under force. Non-covalent handles have already been extensively studied and application is straightforward due to their commercial availability. However, these exclude the possibility of studying extremely mechanically stable proteins and non-covalent handles have shown multiple possible force-loading geometries. Stronger linkages are found for covalent handles using peptide-tags, although the attachment of the linkers is often limited to the termini of the protein and may affect the function of the protein due to its size.

Covalent strategies involving smaller conjugation moieties include lysines, cysteines and non-canonical amino acids. The natural abundance of lysine and cysteine complicates the bioconjugation strategy as a general method. On the other hand, non-canonical amino acids can selectively be targeted and can be conjugated in an efficient manner with several biorthogonal reactions. They provide a powerful tool to introduce linkages due to their size, high rupture force, and selectivity. Introduction of these novel sites provides an opportunity to broaden the possibilities of multiple orthogonal reactions on a single protein. This is particularly useful when combining multiple techniques, for instance the simultaneous FRET measurements with optical tweezers. Implementation of this promising strategy is still limited due to the low incorporation and high costs. Therefore, optimization is still required before it can make its entry as the standard strategy for bioconjugation for single-molecule force spectroscopy. Introducing cysteines in the POI remains the most convenient approach for attaching molecular handles both when high and low forces need to be applied. However, ncAAs are a promising alternative if the POI contains many native cysteine residues or if they play a crucial role in the protein’s function and stability.

## Figures and Tables

**Figure 1 nanomaterials-11-02424-f001:**
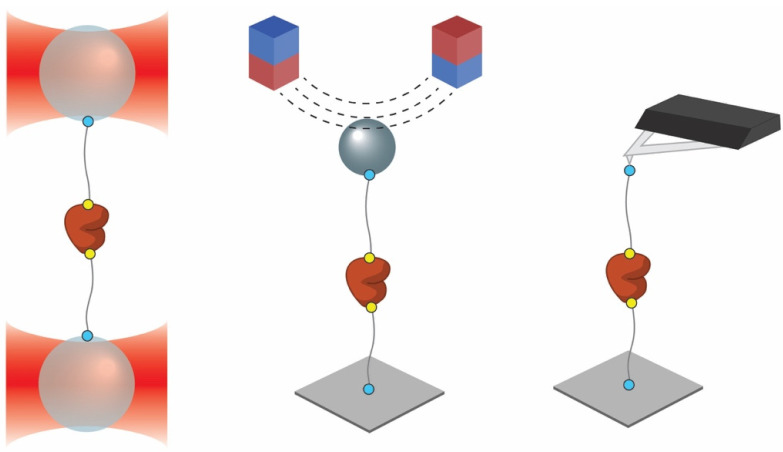
Single molecule force experiments. Examples of configurations for single molecule force experiments on proteins with optical tweezers (**left**), magnetic tweezers (**middle**) and atomic force microscopy (**right**). The protein is ligated to a surface or trapped probe with a flexible linker. Attachment strategies are therefore required between the linker and surface/probe (blue circles) and between the linker and the protein (yellow circles).

**Figure 2 nanomaterials-11-02424-f002:**
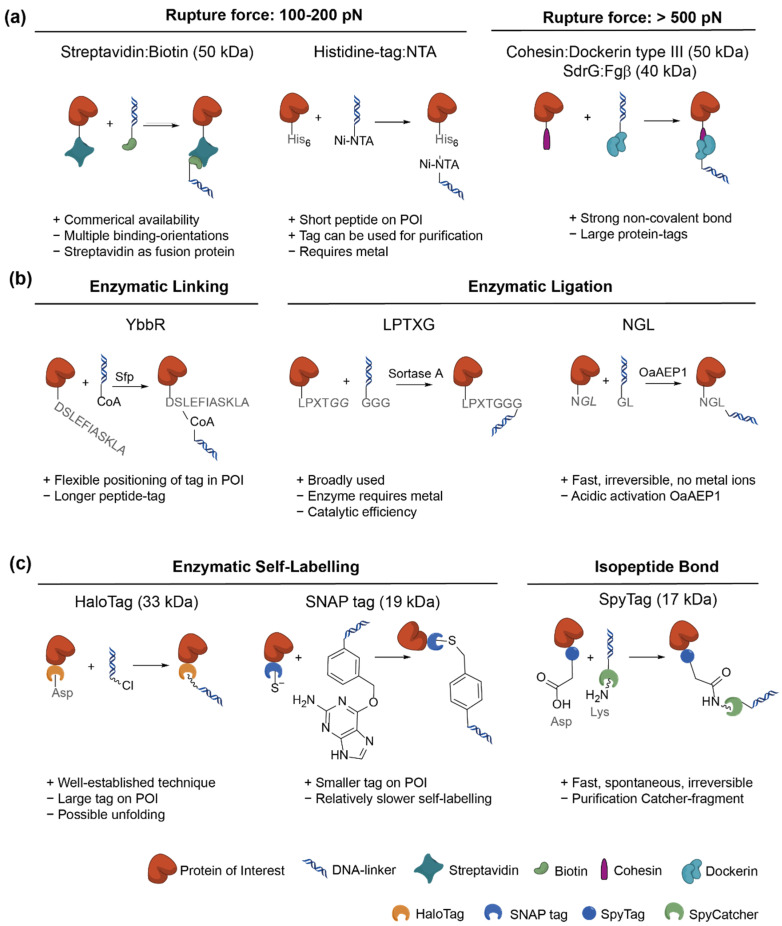
Non-covalent molecular handles and covalent peptide tags for ligation. A comparison of (**a**) non-covalent conjugation strategies, (**b**) enzymatic linking and ligation catalyzed by Sfp phosphopantetheinyl transferase, sortase A or the asparaginyl endopeptidase OaAEP1, and (**c**) enzymatic self-labelling and isopeptide bonds. Amino acid residues are shown in gray and cleaved amino acids are shown in italic.

**Table 1 nanomaterials-11-02424-t001:** Comparison of the labelling conditions of covalent peptide-tags. N-terminal labelling with LPTXG-GGG or NGL-GL tags can be achieved by exchanging the peptide-tags of the POI and DNA-linker. The catalytic efficiency is given by k_cat_/K_m_ and is only a rough indication of the apparent rate constant, as different labels and conditions are used for each tag.

Tag	Labelling Position	Enzyme	Catalytic Efficiency (M^−1^/s^−1^)	Reversible?	Modified Oligonucleotides Available?	Special Experimental Conditions
YbbR-CoA	N-, C-terminus or flexible loops	Sfp	~10^4^ [28]	No	Yes	–
LPTXG-GGG	N- or C-terminus (C-terminal LPTXG)	Sortase A	~10^2^ [40]	Yes	No	Ca^2+^ required [31]
NGL-GL	N- or C-terminus (C-terminal NGL)	OaAEP1	~10^3^ [41]	No	No	Acidic activation OaAEP1 (pH 4) [34]
HaloTag	N- or C-terminus	–	~10^6^ [35]	No	No	–
SNAP tag	N- or C-terminus	–	~10^4^ [42]	No	Yes	1–5 mM DTT recommended [39]
SpyTag	N- or C-terminus	–	~10^3^ [43]	No	No	–

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
