# Peer review of "Bioconjugation Strategies for Connecting Proteins to DNA-Linkers for Single-Molecule Force-Based Experiments"

_nanomaterials, 2021, doi:10.3390/nano11092424_

Round 1

Reviewer 1 Report

Dear authors,

with your review on bioconjugation methods for single molecules you prepared a nice summary on general labeling techniques for proteins. However, there is a number of reviews available already that summarize these techniques in a more comprehensive way than it has been done in your publication. You start with stating that you intend to use DNA-linkers for connecting proteins with probes like an AFM-cantilever or optical tweezers. However, throughout your review you never talk about the connection to the probe, only to the protein intself. DNA itself doesn't play a role either, so altogether it seems to me, as if you could have written this review for any kind of label you aim to put on a protein of interest. So my suggestion is to rewrite the review and more strongly focus on the DNA but also other linkers, how they can be connected to both the protein and the probe and how this affects the force measurements in the end. Other comments I have written directly into the pdf. 

Author Response

We thank both of the reviewers for taking the time to carefully read through our manuscript and for providing us with helpful feedback.

We will address the reviewer’s comments and suggestions point-by-point, referring to line numbers where we have made the relevant changes in the manuscript.

Reviewer 1

General comment – “With your review on bioconjugation methods for single molecules you prepared a nice summary on general labeling techniques for proteins. However, there is a number of reviews available already that summarize these techniques in a more comprehensive way than it has been done in your publication. You start with stating that you intend to use DNA-linkers for connecting proteins with probes like an AFM-cantilever or optical tweezers. However, throughout your review you never talk about the connection to the probe, only to the protein intself. DNA itself doesn't play a role either, so altogether it seems to me, as if you could have written this review for any kind of label you aim to put on a protein of interest. So my suggestion is to rewrite the review and more strongly focus on the DNA but also other linkers, how they can be connected to both the protein and the probe and how this affects the force measurements in the end”

The focus of this short review is to summarise the methods for labelling proteins with a focus on attachments which are suitable for single molecule force spectroscopy measurements using the atomic force microscope, magnetic tweezers or optical tweezers. We therefore focus on details such as the force responses of different labelling strategies. To the best of our knowledge, we are not aware of any other reviews covering this topic specifically for protein-DNA conjugation. Nevertheless, we are happy to add some information into the review about strategies for coupling linkers to probes. We have done so on lines 64 – 68. The text reads: “To conjugate the DNA-linker to the appropriate probe for OT measurements, silica or polystyrene beads coated with for instance (strept)avidin, anti-digoxigenin or azide-groups and biotinylated, digoxigenin- or alkyne- modified DNA primers are commercially available (e.g. from Spherotech, Lumicks, Biomers, and Jena Bioscience).”

  1. Specify in the title, that your review is about tagging proteins with DNA

We have addressed this by changing the title to “Bioconjugation strategies for connecting proteins to DNA-linkers for single-molecule force-based experiments” (lines 2 – 3).

  1. Can you clarify, what is meant by short range interactions? Interactions between the cantilever and the POI? Maybe you can illustrate this a bit better, because to me it actually sounds counter intuitive, as you want to have some measurable tension between the probe and the POI and no flexibility, but apparently only at a certain distance of both to each other?

We have changed the text (lines 31 – 33) to clarify this point. The text now reads: “Background signals can be reduced by excluding non-specific short-range interactions between a surface and the protein of interest (POI) using a flexible linker between the protein and the trapped probe [5].”

  1. What is still not clear to me after reading the text, is how DNA is attached to the probe? I totally get the attachment strategy to the POI, and these strategies are common for attaching any kind of chemical modification to a protein, but what is the interaction or attachment to the probe so that the SMFS measurement can work in the end. Maybe you can extend your review a bit in this directions as this would make the benefit of DNA-linkers vs. PEG-linkers clearer.

As stated above in the response to the ‘general comment’, we have added some additional information about how the DNA is attached to the probe on lines 64 – 68.

  1. How does this relate to the above mentioned 65 pN of the DNA overstretching regime, where you state that this is the limit of applied forces? Or is this value only valid for DNA of 500 bp but usually you use larger fragments?

We thank the reviewer for raising this point, it is important to clarify. In order to address this, we have added the following text on lines 99 – 102 : “Extremely mechanically stable proteins may exceed a rupture force of 500 pN. In such experiments, double-stranded DNA is typically replaced with PEG-based spacers, having the additional benefit of enabling DNA-interacting enzymes to be studied [22].”

  1. You could definitely go a little bit more into detail on this. How is the DNA attached to the protein itself, where is cohesin attached, where dockerin...

We have added more detail from the reference on lines 102 – 138. The text reads: “Studies of the unfolding of these mechanically stable proteins by SMFS would be very challenging with the aforementioned non-covalent handles due to this high rupture force. Cohesin:Dockerin type III and SdrG:Fgß are conjugation pairs that show extreme me-chanical stability despite their non-covalent interactions with rupture forces exceeding 500 pN [22,23] (Figure 2A). Cohesin and Fgβ can be expressed as fusion proteins to the POI, although they are relatively large complexes and may affect the expression, solubility and functioning of the POI. Although the two complexes provide a promising ligation strategy with extraordinary strong non-covalent interactions, in practice it is a less elegant method; an additional method for the ligation of a protein (Dockerin or SdrG) to a DNA-linker is still required.”

  1. Relating to Figure 2:
    1. Clarify abbreviations in figure caption
    2. NGL - abbreviation? Amino acid sequence? Why is NGL italic? Maybe only the cleaved amino acids should be italized, the others normal font.

We have addressed these points by adding a key to the bottom of the figure, and adding to the figure caption, which now states “Figure 2: Non-covalent molecular handles and covalent peptide-tags for ligation. A comparison of (a) non-covalent conjugation strategies, (b) enzymatic linking and ligation catalyzed by Sfp phosphopantetheinyl transferase, sortase A or the asparaginyl endopeptidase OaAEP1, and (c) enzymatic self-labelling and isopeptide bonds. Amino acid residues are shown in gray and cleaved amino acids are shown in italic” (lines 102 – 105).

  1. Make clear, what problems exactly arise. From the example you give, it is not clear to me, but actually illustrates that it works nicely.

We have added text to clarify this. The text now reads: “For instance, while the SpyTag is fused with the POI, the SpyCatcher fragment has been attached to the DNA-linker using an additional linker-protein: the maltose binding protein. This extra protein-fragment is required for convenient purification [39].”

  1. Join this sentence to the sentence before.

We have now done this (lines 167 – 169).

  1. Points relating to Figure 3:
    1. This should be c, I guess?
    2. This ring is not the result of your educts. The DNA should be located at a nitrogen atom and the POI at a carbon atom.
    3. Figure extending over site format
    4. Hydrolytically
    5. Again, nitrogen and triphenylphosphine oxide are eliminated making this thing traceless.

We thank the reviewer for pointing out these important elements of our figure and have made all of the suggested modifications to Figure 3.

Reviewer 2 Report

Comments are listed in the attached file.

Author Response

We thank both of the reviewers for taking the time to carefully read through our manuscript and for providing us with helpful feedback.

We will address the reviewer’s comments and suggestions point-by-point, referring to line numbers where we have made the relevant changes in the manuscript.

Reviewer 2

  1. In lines 112 – 114, it is hard to understand the sentences that explains the bioconjugation technique using isopeptides, especially “a peptide and protein pair”

We agree with the reviewer and have extended this text with the aim of making it clearer (not on lines 128 – 131), as follows: “Proteins containing isopeptide bonds have been engineered by splitting the full protein into two fragments, consisting of a peptide tag and a protein binding partner. This pair can rapidly and spontaneously form an irreversible bond. Examples are the SpyTag:SpyCatcher [39–41] and SnoopTag:SnoopCatcher [42] systems (Figure 2C).”

  1. When starting new line, several lines have indentation, but others are not. It should be corrected.

We have addressed this throughout the manuscript.

  1. In figure 3, (d) should be replaced by (c)

We have made this change

  1. The authors use both “aforementioned” and “afore mentioned”, “straight forward” and “straight-forward”. It should be corrected.

We have made this change throughout the manuscript.

Round 2

Reviewer 1 Report

The authors have responded to the issues I have raised appropriately. I recommend for publication.